# Beneficial Effects of Flaxseed and/or Mulberry Extracts Supplementation in Ovariectomized Wistar Rats

**DOI:** 10.3390/nu14153238

**Published:** 2022-08-08

**Authors:** Jéssica Petrine Castro Pereira, Erika Aparecida Oliveira, Fernanda Aparecida Castro Pereira, Josilene Nascimento Seixas, Camila Souza de Oliveira Guimaraes, Bruno Del Bianco Borges

**Affiliations:** 1Department of Medicine, Federal University of Lavras, Lavras 37200-000, MG, Brazil; 2Department of Biology, Federal University of Lavras, Lavras 37200-000, MG, Brazil

**Keywords:** phytoestrogens, antioxidants, estrogen, compound phenolics, metabolism, body weight

## Abstract

Low endogenous estrogen action causes several injuries. Medicinal plants, such as flaxseed and mulberry, contain substances that have been shown to be effective to the organism. The aim was to verify the effects of flaxseed and/or mulberry extracts on ovariectomized Wistar rats. The animals received supplements of extracts and estrogen or saline by gavage for 60 days and were weighed weekly. Vaginal wash, blood, pituitary, uterus, liver, and kidneys were collected. Phenolic compounds and the antioxidant activity of the extracts, lipid profile, uric acid, liver enzymes, and pituitary weight were measured. Histomorphometric for uterine wall and histopathological analyses for liver and kidney were performed. Flaxseed and mulberry extracts showed great antioxidant activity and large amounts of phenolic compounds. The treatment with extracts had less weight gain, increased pituitary weight, the predominance of vaginal epithelial cells, and reduced TC, LDL-c and lipase activity, similar to estrogen animals. Estrogen or flaxseed + mulberry animals reduced VLDL-c and TAG. HDL-c, uric acid, and liver enzymes did not differ. Estrogen or extracts demonstrated trophic action on the endometrial thickness and have not shown hepatotoxicity or nephrotoxicity. We suggested the beneficial effects of flaxseed and mulberry extract as an alternative to reduce and/or prevent the negative effects caused by low estrogenic action.

## 1. Introduction

Estrogen hormones, especially 17β-estradiol, act on different tissues, such as breast, adipose, vaginal epithelium, uterine wall, and bone [1]; they are involved in the control of food intake, metabolism, the cardiovascular system, lipid profile, among other systems [2,3]. The reduction in estrogenic action, caused by the reduction in plasma estrogen levels or blocking its receptors [4], promotes several symptoms in the organism, such as those that can be observed in menopause symptoms [5,6].

Synthetic hormone replacement therapy has become the first choice of treatment for women seeking to minimize the discomfort caused by a lack of estrogenic action [7]. However, there are contestations and restrictions on their use, especially in women with a family history of breast and uterine cancer [8]. This has led to the search for natural sources in complementary and/or alternative medicine, such as herbal medicines, rich in substances capable of mitigating the effects of the lack of estrogenic action [9].

Medicinal plants have several types of components, including phenolic compounds such as phytoestrogens, which constitute a group of non-steroidal compounds that are known to induce biological responses and mimic and/or modulate estrogenic action [10]. Most of these phytoestrogens have a phenolic ring in their structure, responsible for the ability to bind to hormone receptors, and can act as estrogen agonists or antagonists, depending on the site of action [11]. Different types of medicinal plants contain high concentrations of phytoestrogens [12]. Moreover, medicinal plants contain several compounds with an antioxidant capacity [13].

There are different types of medicinal plants that have phytoestrogens and have effects on several organism systems [12]. Among these plants, flaxseed and mulberry are natural products that contain large amounts of phytoestrogens [14]. Flaxseed (*Linum usitatissimum* L.) has an important source of phenolic compounds and vitamins A, D, E, and K [15]. In addition, flaxseed is rich in lignans, a kind of phytoestrogen commonly consumed in the human diet, and secoisolariciresinol diglycoside (SDG) is the main lignan found [16]. After its metabolism in the intestine and colon, SDG is metabolized into three molecules: secoisolariciresinol (SECO), enterodiol (ED), and enterolactone (EL), which have shown beneficial health effects [17,18]; such bioactive compounds provide greater value for the health of animals and humans due to several factors such as their anti-inflammatory action, antioxidant capacity, and lipid-modulating properties [19]. Due to its properties, flaxseed has been indicated for the reduction of climacteric syndrome symptoms [20].

Mulberry (*Morus nigra*) is a medicinal plant that is widely consumed by the population [21]. The genus Morus is known to contain a variety of phenolic compounds, including isoprenylated flavonoids, coumarins, chromones, xanthones, and phytoalexins [22]. Moroever, they contain several active principles with therapeutic activities, such as an antioxidant capacity, antinociceptive, hypoglycemic, and anti-inflammatory activity, among others [23,24]. Caffeic, chlorogenic, gallic acids, quercetin, and rutin flavonoids were found in the leaf extracts of *Morus nigra* [25]. Additionally, *Morus nigra* fruit extract has a protective action against peroxidative damage of biomembranes and biomolecules [26]. The fruits, bark, stems, and leaves are widely used in folk medicine for therapeutic purposes, such as treating diabetes, hypercholesterolaemia, menopause symptoms, and obesity [24].

Products such as flaxseed and mulberry have preventive and/or curative effects on physiological disorders [12] due to their non-enzymatic antioxidant action [27,28] and ability to mimic and/or modulate estrogenic action [29].

Based on the data presented above, the lack of estrogen action in the body, as occurs in menopause, causes several damages and affects the quality of life mainly in the female organism [30]. Thus, it is necessary to understand the effects of certain medicinal plants rich in phenolic compounds that have antioxidant effects [13] to enable the minimization of the effects caused by the reduction in estrogen action without demonstrating toxicity to the organism.

So, the aim of the present work was to analyze the antioxidant activity and total phenolic compounds in the flaxseed or mulberry extracts, as well as to verify the effects of supplementation with these extracts used on the reproductive system, lipid metabolism, pituitary weight, body weight, liver, and kidney of animals without estrogenic action.

## 2. Materials and Methods

### 2.1. Obtaining and Preparing the Administered Solutions

#### 2.1.1. Flaxseed and Mulberry

The brown flaxseed extract was obtained using the methodology of Galvão et al. (2008) [31], and the mulberry extract by the methodology of Fu et al. (2012), using the dried leaves, stems, and petioles of mulberry [32]. The extracts were protected from light and stored at 4 °C.

For the administration of the flaxseed extract [33] and mulberry [34], doses of 400 mg/kg/day were used for both extracts. For the flaxseed + mulberry group, a solution containing a dose of 200 mg/kg/day of flaxseed plus 200 mg/kg/day of mulberry was used.

#### 2.1.2. Estrogen

Estrogen (estriol—Ovestrion, Eurofama Laboratório S.A.) was macerated, diluted in 500 mL of distilled water, protected from light, and stored at 4 °C. The estrogen was administered at a dose of 0.158 mg/kg [35].

### 2.2. In vitro Antioxidant Assays

#### 2.2.1. Determination of Total Antioxidant Activity by Capturing the Free Radical DPPH (2,2′-Diphenyl-1-Picryl-Hydrazil)

The antioxidant activity was determined using DPPH as a free radical, using the protocol of Rufino et al. (2007). A DPPH curve was performed using an initial DPPH solution (60 µM). For each extract, 100 µL were placed in test tubes containing 3.9 mL of the DPPH radical and homogenized. For the control solution (40 µL of methyl alcohol, 40 µL of acetone, and 20 µL of water), was mixed with 3.9 mL of the DPPH radical. Methyl alcohol was used as white to calibrate the spectrophotometer. The readings (515 nm) were monitored every minute until stabilized. The calculation of the total antioxidant activity consisted of replacing the absorbance equivalent to 50% of the DPPH concentration and finding the result that corresponds to the sample needed to reduce by 50% the initial concentration of the DPPH radical [x = EC50 (mg/L)] [36].

#### 2.2.2. Determination of Antioxidant Activity by the β-Carotene/Linoleic Acid System

For this method, 2 mg of Trolox (6-hydroxy-2,5,7,8-tetramethylchroman-2-carboxylic acid) was dissolved in 5 mL of 70% alcohol and added to 5 mL of ethyl alcohol. Then, 500 mL of distilled water was bubbled with oxygen (oxygenator) for 30 min. Then, 1 mL of chloroform was added to 20 mg of β-carotene, homogenized, and protected from light. The system solution was prepared to contain 40 µL of linoleic acid, 530 µL of Tween 40, 50 µL of the β-carotene solution, and 1 mL of chloroform, homogenized, and the chloroform was placed to evaporate, with the support of the oxygenator, according to the methods of Rufino et al. (2006). The extracts (400 µL) were placed in test tubes containing 5 mL of the system solution, homogenized, and kept in a water bath at 40 °C. The first reading (470 nm) was performed 2 min after homogenization, with other readings performed every 15 min until completion at 120 min. The results are expressed as a percentage of oxidation inhibition [37].

#### 2.2.3. Quantification of the Total Phenolic Content by the Folin-Ciocalteu Method

The method consists of a colourimetric method based on the reduction of the Folin–Ciocalteu reagent by phenolics [38]. For this, 200 μL of each extract, 600 μL of 70% ethanol, 400 μL of Folin–Ciocalteau (Merck), and 2000 μL of the sodium carbonate solution (20% *w*/*v*) were homogenized and 800 μL of the sodium carbonate solution (20% *w*/*v*) was added. Subsequently, the samples were stored for 2 h at room temperature and in a dark ambiance. The absorbance was quantified using a UV-visible spectrophotometer at 735 nm. Gallic acid was used as a standard. For quantification, the phenolic content was calculated from the standard curve of gallic acid in 7 different concentrations, between 0.100 and 0.600 μg/mL [39]. The total phenolic content of the extracts was expressed in GAE (gallic acid equivalents) per 100 g of extract.

### 2.3. Animals

Adult female Wistar rats (*Rattus norvegicus*, *n* = 33) weighing 210 g ± 10 were used and maintained in a 12/12-h photoperiod (7 am to 7 pm) at a temperature of approximately 22 °C, with water and feed ad libitum. The experimental protocol and animal handling were approved by the Ethics Committee of Animal Use of the Federal University of Lavras (UFLA), protocol no. 013/17.

#### 2.3.1. Bilateral Ovariectomy (OVX)

The animals were anaesthetized with ketamine (90 mg/kg B.W., Ceva Santé Animale, Paulínia-SP, Brazil) and xylazine (10 mg/kg B.W., Syntec, Cotia-SP, Brazil). They received a bilateral incision wherein the ovaries were removed, and after the ovariectomy, the incisions were sutured. The animals received a prophylactic dose of veterinary pentabiotic (0.2 mL/rat, Zoetis, São Paulo-SP, Brazil) and analgesic flunixin (meglumine) (2.5 mg/kg B.W., Banamine, Chemitec Agro-Vetrinária, São Paulo-SP, Brazil).

#### 2.3.2. Experimental Groups

The animals were divided into 5 groups and treated with: (a) saline (*n* = 6), (b) estrogen (*n* = 6), (c) flaxseed extract (*n* = 7), (d) mulberry extract (*n* = 7), and (e) a mixture of flaxseed plus mulberry extract (*n* = 7).

### 2.4. Experimental Protocol

#### 2.4.1. Solution Administration

After 15 days of estrogen depletion and surgical recovery, all of the groups received orogastric treatments every day for 60 days. The volume was approximately 500 µL/animal through the orogastric gavage procedure [40].

#### 2.4.2. Body Weight

Each week, during all treatments, the animals were weighed with a precision electronic digital scale SF-400.

#### 2.4.3. Euthanasia

After treatment, the animals were anaesthetized with ketamine (90 mg/kg B.W, Ceva Santé Animale, Paulínia-SP, Brazil) and xylazine (10 mg/kg B.W., Syntec, Cotia-SP, Brazil). The blood samples were obtained through a cardiac puncture, and the liver, kidney, uterus, and pituitary gland were extracted and processed for correct storage.

#### 2.4.4. Vaginal Wash

On the day of euthanasia, the animals were subjected to vaginal lavage to view the vaginal epithelial cells under an optical microscope (Olympus CX22 RFS2) with 10× and 40× objective lenses [41].

#### 2.4.5. Pituitary Weight

The pituitary was dried on paper towels and, using an analytical balance with a sensitivity of 0.0001 g, model PA214P (OHAUS), the pituitary weight was obtained.

### 2.5. Biochemical Analyses

The blood samples from all of the animals were centrifuged (3500 rpm/20 min), and the plasma was collected and stored in a freezer (−80 °C).

#### 2.5.1. Quantification of Total Cholesterol (TC), Triacylglycerols (TAG), Very-Low-Density Lipoprotein Cholesterol (VLDL-c), Low-Density Lipoprotein Cholesterol (LDL-c), and High-Density Lipoprotein Cholesterol (HDL-c):

The concentrations of TC and TAG were determined using enzymatic kits (BioTécnica, Varginha-MG, Brazil), as described in the protocol. The solutions containing the standard, sample, and the blank solution were incubated at 37 °C for 10 min and analyzed using a spectrophotometer at 505 nm. VLDL-c and LDL-c were calculated using the Friedewald formula [42]. For the quantification of HDL-c, a reagent kit (Bioclin K015, Belo Horizonte-MG, Brazil) was used according to the manufacturer’s protocol. The solutions were prepared, homogenized, incubated in a water bath at 37 °C for 5 min and analyzed using a spectrophotometer at 500 nm. HDL-c was calculated from the equation contained in the manufacturer’s protocol.

#### 2.5.2. Quantification of Lipase, Uric Acid, Glutamic-Oxalacetic Transaminase (GOT) and Glutamic-Pyruvic Transaminase (GPT)

The concentrations of lipase, uric acid, GOT, and GPT were analyzed using specific reagent kits according to the manufacturer’s protocol, respectively. To quantify the plasma lipase activity (BioTécnica, Varginha-MG, Brazil), the temperature of the photometer was adjusted to 37 °C at 580 nm. All of the solutions were prepared, and absorbances were recorded at 90 s (A1) and 180 s (A2). To determine the uric acid concentration (BioTécnica, Varginha-MG, Brazil), the solutions were prepared, homogenized, and incubated at 37 °C for 10 min. Absorbance was analyzed on a spectrophotometer at 505 nm. To quantify the liver enzymes, GOT and GPT (Bioclin K034 and K035, respectively, Belo Horizonte-MG, Brazil) calibration curves were made for both. Absorbances were determined at 505 nm.

All of the analyses were calculated from the equation contained in the respective manufacturer’s protocol.

### 2.6. Histological Analysis

#### 2.6.1. Collection and Processing of the Uterus and Liver

The uterus, liver, and kidney were collected and sliced in saline. Subsequently, the material was fixed in a 10% (*v*/*v*) formalin solution for at least 48 h.

#### 2.6.2. Histological Procedures

After fixation, the samples were placed on cassettes, submitted to the histotechnician for dehydration in a gradual series of ethyl alcohol (70, 80, 90, 95, and 100% (*v*/*v*), for 20 min each), diaphanized in xylol, and soaked in paraffin at 60 °C (two baths of 40 min each). The paraffin blocks containing the samples were sectioned in microtome at a thickness of 4 µm.

The sections of the uterus, liver, and kidney were dewaxed, hydrated, and stained with Harris haematoxylin for 2 min. Subsequently, they were washed in running water and stained with eosin for 4 min. Finally, the cuts were again dehydrated, diaphanized, and affixed using synthetic Canada balm, and a coverslip was added (adapted from Junqueira, 1983).

Another kidney slice was stained with periodic acid-Schiff (PAS). The slice was dewaxed, hydrated, and oxidized in 0.5% (*v*/*v*) periodic acid for 15 min. Then, it was washed in running water for 5 min, subjected to staining with Schiff’s reagent for 30 min, and was washed again and stained with Harris Haematoxylin.

#### 2.6.3. Histomorphometric Analysis of the Uterus

The middle third of the uterine horns was analyzed, and six histological sections were made for each animal. The sections were analyzed using a photomicroscope (Olympus CX22 RFS2) with a digital camera (SC30) with a 10× objective lens. The measurement of the endometrium was obtained by the distance from the apical surface of the luminal cells to the limit of the endometrium with the myometrium [43]. These analyses were performed using the image analysis system ImageJ, version 4.5.0.29 (National Institutes of Health, Bethesda-Maryland, USA).

#### 2.6.4. Histopathological Analysis of the Liver

The slides were examined using the histological scoring system for non-alcoholic fatty liver disease (NAFLD) adaptation [44,45].

The samples were analyzed for the presence of steatosis, ballooning of liver cells, infiltration of inflammatory cells, necrosis, congestion of vessels, and fibrosis, among other findings. For that, scores were established to classify the results, based on the type of injury and the intensity of the injuries as follows: (1) steatosis: 0 (<5%), 1 (5–33%), 2 (33–66%) and 4 (>66%), (2) inflammation: 0 (no focus), 1 (2–4 foci per 10 × field), 2 (4–8 foci per 10 × field), 3 (more than 8 foci per 10 × field), (3) fibrosis: 0 (without fibrosis), 1 (perisinusoidal or periportal), 2 (perisinusoidal and periportal), 3 (bridged), 4 (cirrhosis), and (4) core/cytoplasm ratio (ballooning): 0 (≤1:2), 1 (1:2 to 1:3), 2 (1:3 to 1:4), and 3 (≥1:4). Finally, the scores were added to verify the overlap of lesions.

#### 2.6.5. Histopathological Analysis of the Kidney

To check for possible morphological changes in the renal parenchyma, histological sections were observed under a light microscope (OLYMPUS, CX22LED) and evaluated for increases in the urinary space area, presence of epithelial cells exfoliated in the tubular lumen, vascular congestion, perivascular oedema, intratubular hyaline deposits, tubular vacuolization, loss of the brush border, tubular dilation, and leukocyte infiltration. The attribution of scores was defined according to established criteria [46]. Considering the percentage of renal parenchyma with changes, the following scoring was applied: the absence of lesions (grade 0), 1–20% (grade 1), 21–40% (grade 2), 41–60% (grade 3), 61–80% (grade 4), and 81–100% (grade 5). The sum of all of the numerical scores in each group was considered the total histopathological score.

### 2.7. Statistical Analysis

The data were subjected to analysis of variance (ANOVA), and the treatment means were grouped by the Scott–Knott test (1974). The analyses were processed by the program R (R CORE TEAM, 2014), and the graphics were generated by the program GraphPad Prism (GraphPad PRISM 5).

## 3. Results

### 3.1. Total Antioxidant and Total Phenolic Compounds Analysis

The percentage of inhibition of the flaxseed extract in sequestering the DPPH radical was 74.5%, and that of the mulberry extract was 73.4%. The mulberry extract alone showed 44.8% antioxidant activity by the beta-carotene/linoleic acid system, while the flaxseed extract could not be quantified (Table 1). The mulberry extract had a total phenolic content of 1482.6 ± 37 mg GAE/100 g of extract, and the flaxseed extract had a total phenolic content of 1395.4 ± 11 mg GAE/100 g of extract (Table 1).

### 3.2. Weight Gain

There was less weight gain in the animals treated with estrogen, and extracts of flaxseed, mulberry, and flaxseed + mulberry in relation to saline (F_4,27_ = 3.35, *p* = 0.023) (Figure 1A).

### 3.3. Cells Present in the Vaginal Lavage

The animals that received saline demonstrated a predominance of leukocyte cells. On the other hand, the animals treated with estrogen or with different extracts demonstrated the predominance of epithelial cells, a specific characteristic of estrogenic action in the vaginal epithelium (Figure 2).

### 3.4. Pituitary Weight

The animals treated with estrogen or different extracts demonstrated a greater weight of the pituitary gland compared to the group treated with saline (F_4,27_ = 3.46, *p* = 0.021) (Figure 1B).

### 3.5. Biochemical Analysis

In the lipid profile, the animals with estrogenic action and those who received extracts showed a reduction in TC concentration; moreover, the animals treated with flaxseed + mulberry showed a greater reduction than among the animals treated with the extracts (F_4,27_ = 31.3, *p* < 0.001) (Figure 3A).

For TAG and VLDL-c, the flaxseed + mulberry and estrogen groups showed significant reductions (Figure 3B,C). However, the animals treated with flaxseed or mulberry extract did not show any significant difference. In addition, animals with estrogenic action or supplemented with extracts demonstrated significant reduction in LDL-c concentrations (F_4,27_ = 28.34, *p* < 0.001) (Figure 3D). However, in relation to HDL-c concentrations, no significant differences were observed among the groups (F_4,27_ = 1.36, *p* = 0.274) (Figure 3E). Moreover, a reduction in the lipase activity was observed in the animals treated with estrogen or extracts supplementation (Figure 3F).

According to the uric acid analyses (F_4,27_ = 0.591, *p* = 0.672), GOT (glutamic-oxalacetic transaminase) (F_4,27_ = 2.49, *p* = 0.069) and GPT (glutamic-pyruvic transaminase) (F_4,27_ = 2.31, *p* = 0.082), no significant differences were observed between the different types of treatments (Figure 3G–I).

### 3.6. Endometrial Thickness

The animals with estrogenic action demonstrated greater endometrial thickness than the other groups analyzed. In addition, the animals treated with extracts showed greater endometrial thickness than the saline group (F_4,27_ = 7.84, *p* < 0.001) (Figure 4).

### 3.7. Liver and Kidney Histopathology

The changes commonly observed in cases of non-alcoholic fatty liver disease (NAFLD) were not evidenced; however, the animals without estrogenic action and treated with saline showed a higher number of liver changes, demonstrating that only 33.3% of the animals had a score of 0, and predominance of hepatocytes with fat accumulation, characterizing steatosis, and foci of inflammation (Table 2). These animals demonstrated aggregates of inflammatory cells and a moderate vacuolization of hepatocytes, with a slight distortion of the architecture and a greater number of cells with accumulation of fat droplets (Figure 5A).

No significant changes were observed in the renal tissue of the animals analyzed in the present study (Figure 6 and Table 3).

## 4. Discussion

In the present work, we observed that the flaxseed or mulberry extracts demonstrated a high concentration of phenolic compounds and have high antioxidant power. Moreover, animals without estrogen action treated with these extracts demonstrated less weight gain, a predominance of vaginal epithelial cells in the vaginal lavage, improvement in the lipid profile, increase in endometrial thickness and pituitary weight, and demonstrated low hepatic and renal toxicity, which are results similar to those of animals treated with exogenous estrogen.

The antioxidant activity of fruit and plant extracts is generally related to the presence of phenolic compounds and has been gaining attention over the years [47]. Plant extracts with elevated concentrations of anthocyanins and other flavonoids demonstrated high radical scavenging activity when analyzed by the DPPH method [48]. The ability to sequester the free radical DPPH from brown flaxseed oil was 78.5% [49], a result similar to that found in the present work (Table 1). This can be explained by a large amount of unsaturated fatty acids in flaxseed oil and by the presence of phenolic compounds in flaxseed [50].

Hassimotto and collaborators also found a high inhibition efficiency of DPPH radicals in mulberry (*Rubus eubatus*) cultivars extracted with 80% ethanol [51], which is similar to the results obtained in the present work with mulberry extract (Table 1). The presence of bioactive compounds, such as prenylated phenolics, is characteristic of mulberry leaves and is responsible for the high antioxidant activity [52].

The mulberry extract was subjected to the beta-carotene/linoleic acid test [53,54] and showed oxidative inhibitory activity (44.8%) (Table 1). Hassimoto and collaborators demonstrated that the oxidation of β-carotene was inhibited in all mulberry cultivars, resulting in an inhibition rate that varied between 66.0 and 76.0% [51]. However, by this assay, the antioxidant activity of the flaxseed extract could not be quantified. The antioxidant activity potentially could not have been demonstrated by this method due to the intrinsic characteristics of the seed in relation to the solution in which it was solubilized [31].

Depending on the types of solvents used to extract mulberry, different levels of phenolic compounds are observed. Although the present work used mulberry leaves, petioles, and stems, the total phenolic content of the ethanolic mulberry extract showed a value of 1482 ± 37 mg of GAE/100 g, similar to that observed in the previous study when the extraction was performed with acetone and demonstrated 1022 ± 46 mg of GAE/100 g of phenolic compounds [55].

Regarding the total phenolic compounds of flaxseed, Barroso and collaborators found that the brown and golden flaxseeds showed no significant difference in content at 1332 ± 0.09 mg of GAE/100 g and 1039 ± 0.21 mg of GAE/100 g, respectively [56]. Similar to that observed in this work, brown flaxseed extract demonstrated 1395.4 ± 11.8 mg of GAE/100 g (Table 1). Thus, the high concentrations of total phenolics and high antioxidant power found in the extracts used can be, at least in part, related to the positive results observed in animals without estrogenic action.

Estrogen reduction promoted by ovariectomy results in greater weight gain and adiposity, changes in the lipid profile [57], and uterine atrophy [58]. The administration of 17β-estradiol benzoate (EB) to ovariectomized (ovx) animals was associated with a reduction in the amount of ingested food and less body weight gain [59].

Male rats supplemented with flaxseed flour for 35 days demonstrated less weight gain in relation to the group without supplementation [60]. The chronic administration of mulberry (*Rubus rosifolius*) infusion was able to significantly reduce the Lee Index of animals, corresponding to the BMI (body mass index) in humans [61]. The work demonstrated the action of mulberry and flaxseed extracts in reducing weight gain in animals without estrogenic action (Figure 1A). Thus, the animals supplemented with flaxseed and mulberry demonstrated similar results to the animals treated with estrogen, suggesting a possible estrogenic action or anorexigenic actions of these supplementations.

The cells of the vaginal epithelium respond with great sensitivity to sex steroids [62]. The estrogenic action promotes superficial cells with a pycnotic nucleus, keratinize, become acidophilic, and finally desquamate [63,64].

The treatment with an extract of *Trifolium pratense* L., a red clover rich in isoflavones, in ovx rats demonstrated a distinct pattern of vaginal cells beginning with the leukocyte population, progressing to nucleated cells (middle of treatment), and ending with cornified cells [65]. The flaxseed extract caused changes in the vaginal epithelium of immature ovx rats [66]. The extracts used in the present work demonstrated a predominance of epithelial cells, similar to the animals treated with estrogen, demonstrating possible estrogenic action of extracts in the vaginal epithelium. On the other hand, the saline-treated animals showed a predominance of leukocytes and small cells, similar to hypoestrogenism [63,64] (Figure 2).

Estrogen modifies biological aspects in the pituitary gland [67], including the regulation of lactotrophic and gonadotrophic homeostasis, prolactin synthesis and secretion, and has a trophic effect [68,69]. The specific blockade of estrogen’s mechanism of action on the pituitary reduced the estrogen effects on the development of lactotrophic hyperplasia [68,70]. The animals supplemented with extracts used demonstrated a significant increase in the pituitary weight similar to animals treated with estrogen, suggesting the possible trophic effect of these supplements, such as an estrogenic action (Figure 1B).

Estrogen acts on the liver and increases the synthesis of HDL-c and reduces the synthesis of VLDL-c, among other actions, thus promoting a beneficial balance in the metabolism of these lipoproteins and cardiovascular protection [71]. In the same way, flaxseed has great potential in the prevention of cardiovascular diseases, demonstrating a reduction in TC and LDL-c in studies in humans and animals [72]. The presence of fiber, both in the grain and in the bran, promotes an increase in intestinal transit and consequently decreases the absorption of lipids and cholesterol, which is also responsible for the hypocholesterolemic properties of this oilseed [73,74]. Ovx rats with flaxseed and soy supplementation demonstrated improvement in the lipid profile [75]. Moreover, flaxseed can delay the progression of atherosclerotic lesions [76]. These data may be due to the high concentrations of unsaturated fatty acids present in flaxseed, verifying a more substantial effect in reducing LDL-c, which is considered a cardiovascular risk factor [77].

*Morus nigra* extract reduced TC, TAG, and VLDL-c concentrations [78]. These lipid-lowering effects of *Morus nigra* can be explained by the high concentrations of flavonoids present in this plant [79], similar to terpenoids, which have a lipid-lowering action [80]. Supplementation with *Rubus coreanus* Miquel, known as Korean blackberry, increased plasma HDL-c and decreased TC concentrations [81].

The data of the present work demonstrated that animals without estrogenic action have an elevated lipid profile. However, the supplementation with extracts used had a preventive effect with the improvement in the lipid profile of these animals, similar to animals with estrogenic action (Figure 3A–E). Thus, it is suggested that flaxseed and/or mulberry supplementation demonstrate efficacy in reducing the risk of cardiovascular disease in organisms without estrogenic action.

There is a relationship between estrogen, pancreatic insulin secretion, and lipase activity [82]. The administration of estrogen in diabetic rats demonstrated reduced rates of glucose absorption and an 84% reduction in plasma lipase activity, causing considerable decreases of 28% and 71% in TC and TAG, respectively [83]. The supplementation with extracts used demonstrated a reduction in plasma lipase activity in animals similar to animals with estrogenic action (Figure 3F). Therefore, these data demonstrate one more factor for cardiovascular protection caused by the extracts used.

The rats that received flaxseed supplementation showed a decrease in plasma uric acid levels [84]. On the other hand, the administration of *Morus nigra* leaves for 30 days did not influence the concentration of uric acid [85]. The results did not show differences in plasma uric acid levels between the different groups analyzed (Figure 3G).

Estrogen increases protein synthesis, uterine musculature, the proliferation of stromal and epithelial cells, development of endometrial glands and new blood vessels, and fluid and electrolyte retention, promoting uterine enlargement [63,64]. Figure 4 demonstrates that saline-treated animals had uterine atrophy due to the absence of estrogenic action caused by the removal of the ovaries, and when these animals were treated with estrogen, there was a great increase in the endometrial thickness.

Some plant extracts contain high concentrations of phenolic compounds such as phytoestrogens, which constitute a group of non-steroidal compounds that are known to induce biological responses and mimic and/or modulate estrogenic action [10]. These compounds have long been recognized for their uterotrophic activity in a variety of animal species [86,87]. The aqueous methanolic extract of flaxseed promoted a significant increase in uterine weight and ovarian weight in mice [88]. In sexually immature and non-ovariectomized rats, flaxseed ethanolic extract demonstrated an increase in uterine weight [66]. Uterine growth was observed in animals treated with a dose of 500 mg/kg methanolic extract of *Morus alba* [89]. The extracts used demonstrated a trophic effect on the endometrial thickness (Figure 4). Thus, it is suggested that substances present in flaxseed and mulberry extracts, for example, phenolic compounds such as phytoestrogens [90,91], may interact with estrogen receptors in the endometrium and promote the trophic effects observed.

One of the major problems related to substance supplementation is that supplements may cause liver and kidney damage, so the next step was to verify the liver and kidney integrity of these animals. In the presence of liver necrosis and tissue destruction by toxic agents, liver transaminases (GOT and GPT) are usually found at high levels in the plasma [92]. The content of fatty acids and lignan in flaxseed attenuates the progression of non-alcoholic liver steatosis in chickens and improves liver morphological parameters and serum GOT levels [93]. Flaxseed oil is rich in omega-3 polyunsaturated fatty acids, mainly α-linolenic acid [94]. Supplementation with this oil prevented liver steatosis and insulin resistance in rats [95] and promoted reductions in GOT and GPT, as well as an improvement to the liver damage caused by alcohol, indicating a protective effect against liver damage caused by chronic ethanol [96]. Moreover, rats that received flaxseed oil orally for 60 days also did not show histological changes in liver tissue, with reduced fat deposition in hepatocytes [97], as observed in the results (Figure 5).

The administration of tea from the leaves of *Morus nigra* was considered to have low toxicity; furthermore, this treatment did not generate changes in the plasma levels of GOT and GPT [85].

The animals treated with estrogen or flaxseed + mulberry showed normal tissue architecture and no vacuolization (Figure 5B,E). The treatment with flaxseed extract showed normal tissue architecture (C1) and strands of hepatocytes in a row, with a nucleus:cytoplasm ratio of 1:2. Finally, the animals treated with mulberry extract also demonstrated normal tissue architecture, where hepatocytes with less eosinophilic cytoplasm and few vacuolized cells were observed (Figure 5D). Thus, the animals that received estrogen or extracts did not present liver damage and had a lower deposit of fats in hepatocytes in relation to saline animals, demonstrating that animals without estrogenic action may have an accumulation of fat in the liver and that the extracts used may prevent this accumulation.

Additionally, renal damage needs investigation in supplementation studies. Supplementation with flaxseed oil in rats treated with arsenic demonstrated that flaxseed was able to protect against renal damage, maintain integrity, accelerate the regeneration of injured organelles, strengthen the endogenous antioxidant defense, and neutralize the arsenic toxicity mediated by free radicals [98]. The use of substances from the bark of *Morus alba* demonstrated effectiveness against nephrotoxicity induced by the administration of paracetamol and verified the minimal amount of nephritic cell destruction, confirming the potential nephroprotective effect of *Morus alba* extract [99]. As demonstrated in Figure 6 and Table 3, supplementation with the extracts did not promote toxic effects on the renal tissue, thus indicating the absence of nephrotoxicity in both supplemented extracts.

## 5. Conclusions

The data from the present work demonstrate that the extracts used are rich in phenolic compounds and have high antioxidant power. The supplementation with flaxseed and/or mulberry extracts has many health benefits to the organism and promotes beneficial effects on different tissues and systems of the body without estrogenic action, similar to those found in animals with estrogenic action, without demonstrating toxicity to the organism. Therefore, a nutraceutical alternative and/or effective complement is suggested to reduce and control the negative effects generated by the decrease or absence of estrogenic action in the organism.

## Figures and Tables

**Figure 1 nutrients-14-03238-f001:**
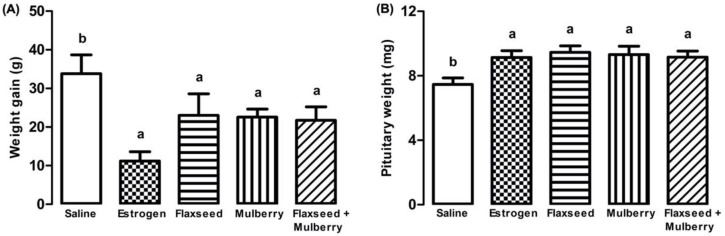
(**A**) Weight gain (grams) at 60 days and (**B**) pituitary weight (milligrams) of animals treated with saline, estrogen, flaxseed extract, mulberry extract, or flaxseed + mulberry. The bars with the same letter do not differ statistically from each other by the Scott–Knott test (*p* < 0.05). Values represent means ± standard deviation.

**Figure 2 nutrients-14-03238-f002:**
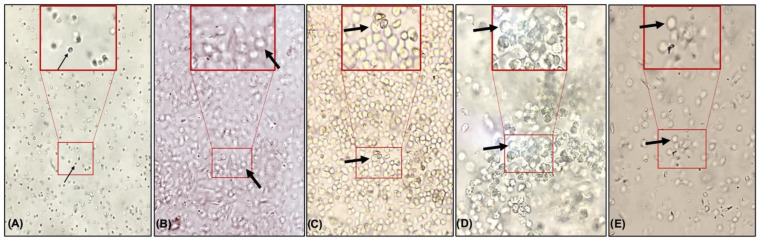
Photomicroscopy (10×) of cells present in the vaginal lavage of animals treated with saline (**A**), estrogen (**B**), flaxseed extract (**C**), mulberry (**D**), and flaxseed + mulberry (**E**). The thin arrow indicates leukocyte cells, and the thick arrow indicates epithelial cells.

**Figure 3 nutrients-14-03238-f003:**
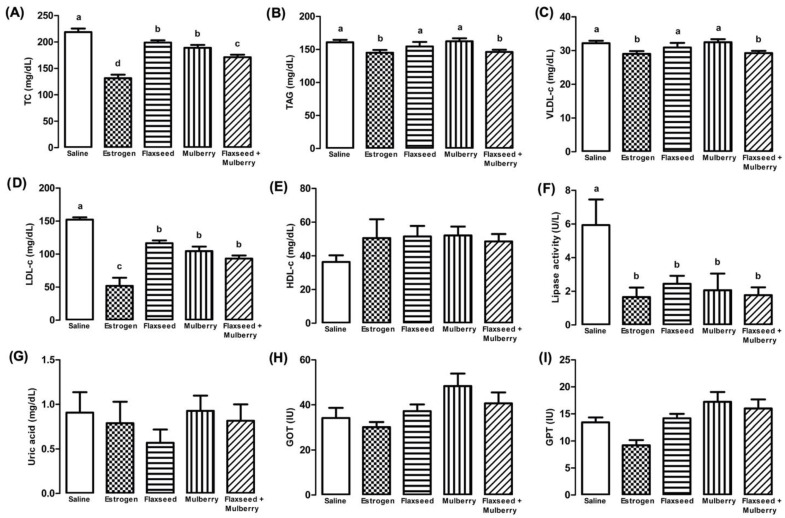
Plasma levels of total cholesterol (TC (mg/dL) (**A**), triacylglycerol (TAG (mg/dL) (**B**), very-low-density lipoprotein cholesterol (VLDL-c (mg/dL) (**C**), low-density lipoprotein cholesterol (LDL-c (mg/dL) (**D**), high-density lipoprotein cholesterol (HDL-c (mg/dL) (**E**), lipase activity (U/L) (**F**), uric acid (mg/dL) (**G**), glutamic-oxaloacetic transaminase (GOT (IU)) (**H**) and glutamic-pyruvic transaminase (GPT (IU)) (**I**) in animals treated with saline, estrogen, flaxseed extract, mulberry extract or flaxseed + mulberry. The bars with the same letter do not differ statistically from each other by the Scott–Knott test (*p* < 0.05). Values represent means ± standard deviation.

**Figure 4 nutrients-14-03238-f004:**
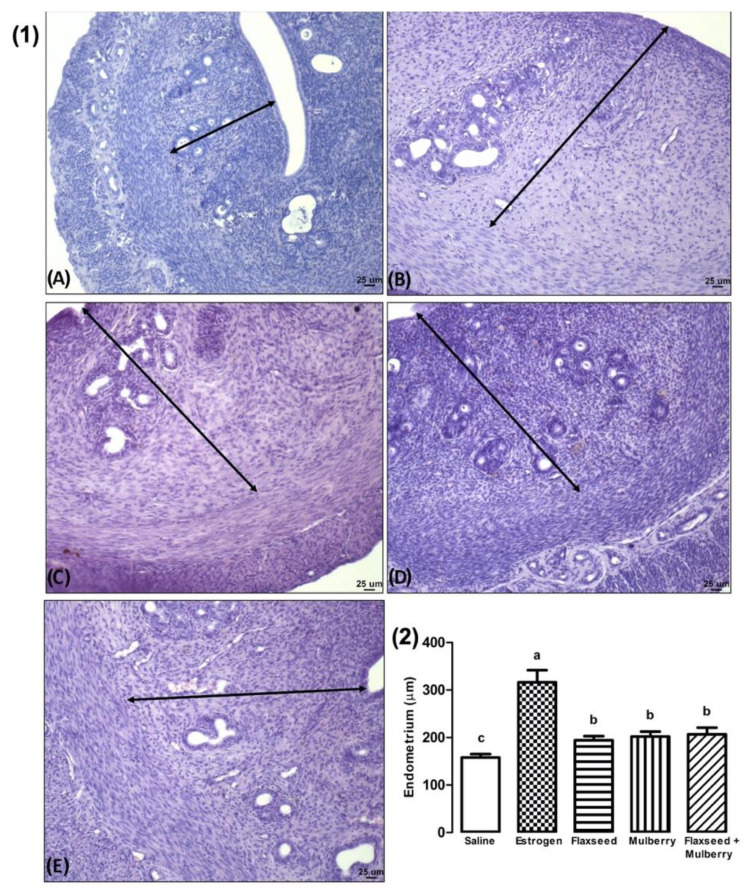
(**1**) Photomicroscopy (HE, 10×) of the endometrium of animals treated with saline (**A**), estrogen (**B**), flaxseed extract (**C**), mulberry extract (**D**), and flaxseed + mulberry (**E**). Arrow length: the thickness of the endometrium (µm). (**2**) Endometrial length (µm) of the different groups analyzed. The bars with the same letter do not differ statistically from each other by the Scott–Knott test (*p* < 0.05). Values represent means ± standard deviation.

**Figure 5 nutrients-14-03238-f005:**
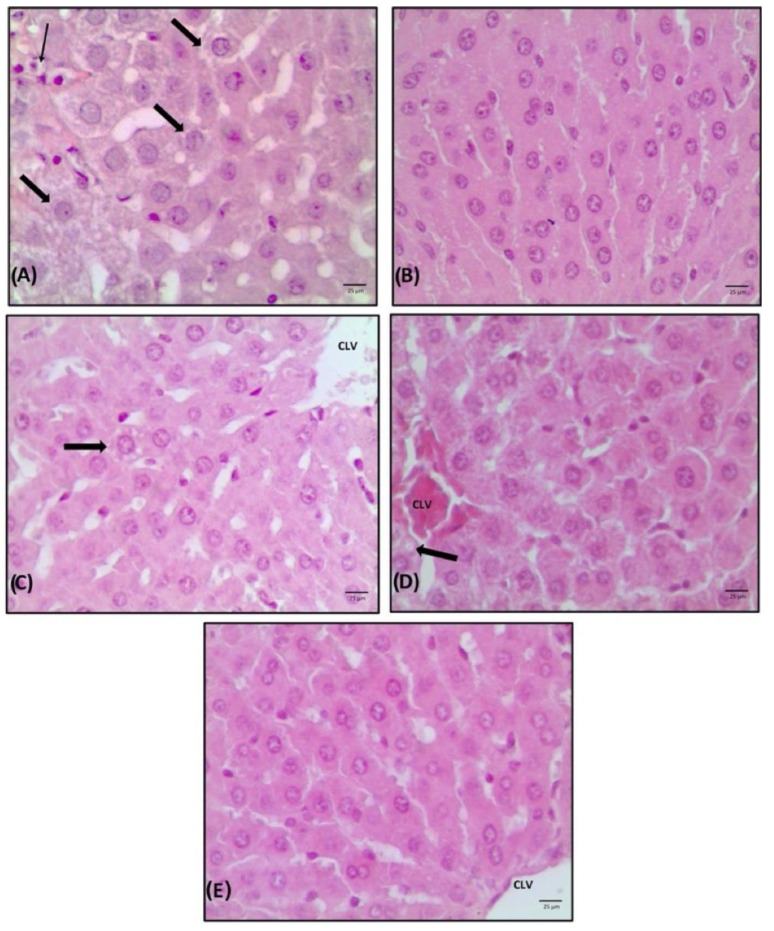
Liver photomicroscopy with haematoxylin-eosin (HE) staining. Increase 20× of animals treated with saline (**A**), estrogen (**B**), flaxseed extract (**C**), mulberry extract (**D**) and flaxseed + mulberry (**E**). The thin arrow indicates inflammatory infiltrate, and the thick arrow indicates steatosis. PS: portal space. CLV: central lobular vein.

**Figure 6 nutrients-14-03238-f006:**
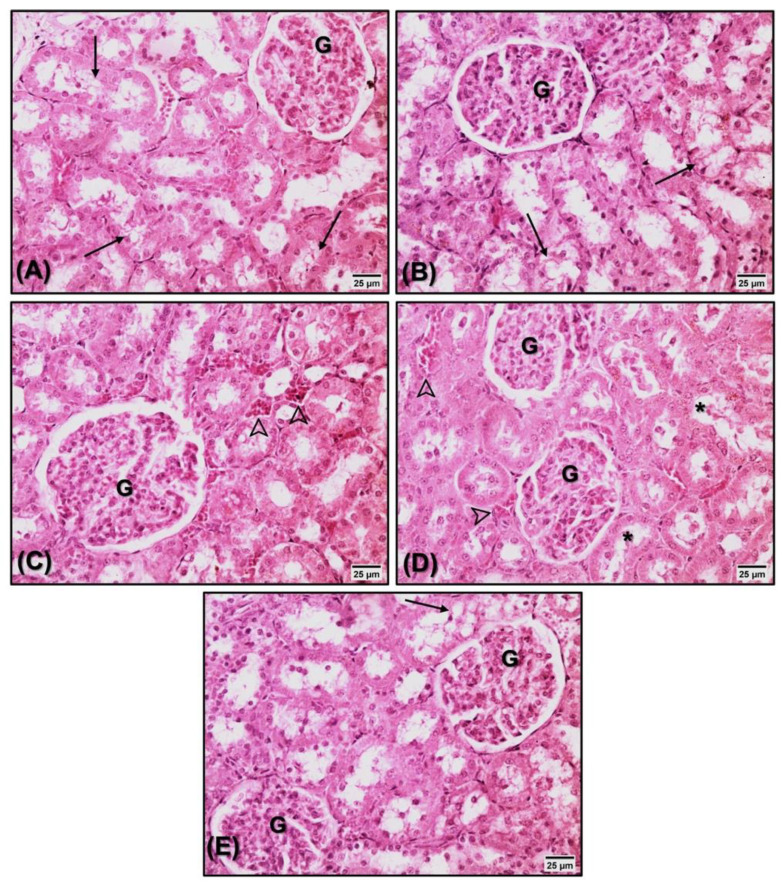
Kidney histopathology (HE, 40×) of animals treated with saline (**A**), estrogen (**B**), flaxseed extract (**C**), mulberry extract (**D**), and flaxseed + mulberry (**E**). Mild morphological alterations, such as tubular vacuolization (arrows), vascular congestion (arrowheads), and tubular dilatation (*). No significant differences were observed between the groups. G: glomerulus.

**Table 1 nutrients-14-03238-t001:** Antioxidant capacity by the percentage of oxidation inhibition (DPPH and β-carotene/linoleic acid) and total phenolic compounds (TPC) of flaxseed and mulberry extracts. IC50: inhibitory concentration (mg/mL). TPC: total phenolic compounds. DPPH: 2,2′-Diphenyl-1-Picryl-Hydrazil. GAE: gallic acid equivalents. The values represent the means ± standard deviation.

	IC50	% Oxidation Inhibition		TPC
Extract	(mg/mL)	DPPH	β-Carotene/Linoleic Acid	(mg GAE/100g)
**Flaxseed**	6.93 ± 0.431	74.55 ± 4.64	-	1395.4 ± 11.83
**Mulberry**	0.04 ± 0.007	73.44 ± 14.1	44.77 ± 28.90	1482.6 ± 37.08

**Table 2 nutrients-14-03238-t002:** Results of the histopathological analyses of the liver of animals treated with saline, estrogen, flaxseed, mulberry, and flaxseed + mulberry extract. Adapted from [1] Kleiner et al. (2005). Steatosis grade: 0 (<5%), 1 (5–33%), 2 (33–66%) and 3 (>66%). Inflammation: 0 (no focus), 1 (2–4 foci per 10X field), 2 (4–8 foci per 10X field), 3 (more than 8 foci per 10X field). Fibrosis stage: 0 (without fibrosis), 1 (perisinusoidal or periportal), 2 (perisinusoidal and periportal), 3 (bridging), 4 (cirrhosis). Core/cytoplasm ratio (ballooning): 0 (≤1:2), 1 (1:2 to 1:3), 2 (1:3 to 1:4) and 3 (≥1:4). Total score: the sum of all scores.

					Experimental Groups		
Histopathologic Damages	Definition	Score	Saline	Estrogen	Flaxseed	Mulberry	Flaxseed + Mulberry
**Steatosis grade**	<5%	0	50%	100%	71.40%	85.70%	100%
	5–33%	1	50%	0	28.50%	14.20%	0
	33–66%	2	0	0	0	0	0
	>66%	3	0	0	0	0	0
**Inflammation**	no focus	0	66.60%	80%	71.40%	85.70%	57.10%
	2–4 foci per 10× field	1	33.30%	20%	14.20%	14.20%	42.80%
	4–8 foci per 10× field	2	0	0	14.20%	0	0
	>8 foci per 10× field	3	0	0	0	0	0
**Fibrosis stage**	no fibrosis	0	100%	100%	100%	100%	100%
	perisinusoidal or periportal	1	0	0	0	0	0
	perisinusoidal and periportal	2	0	0	0	0	0
	bridging	3	0	0	0	0	0
	cirrhosis	4	0	0	0	0	0
**Ballooning**	≤1:2	0	100%	100%	100%	100%	100%
	1:2 a 1:3	1	0	0	0	0	0
	1:3 a 1:4	2	0	0	0	0	0
	≥1:4	3	0	0	0	0	0
**Total Score**		0	33.30%	80%	57.10%	71.40%	57.10%
		1	50%	20%	28.50%	28.50%	42.80%
		2	16.60%	0	0	0	0
		3	0	0	14.20%	0	0

**Table 3 nutrients-14-03238-t003:** Results of the histopathological analyses of the kidneys of animals treated with saline, estrogen, flaxseed, mulberry, and flaxseed + mulberry extract. Adapted from Yarijani et al. (2019). Grade 0: the absence of injuries. Grade 1: 1–20%. Grade 2: 21–40%. Grade 3: 41–60%. Grade 4: 61–80%. Grade 5: 81–100%. Total score: the sum of all scores.

	Experimental Groups				
Histopathologic Damages	Saline	Estrogen	Flaxseed	Mulberry	Flaxseed + Mulberry
**Bowman’s space**	1.3	1.4	1.6	1.0	1.3
**Vascular congestion**	1.5	1.6	2.2	2.3	2.0
**Perivascular edema**	1.0	1.4	1.1	1.4	1.1
**Intra-tubular casts**	0.2	0	0	0	0
**Tubular vacuolization**	1.5	1.6	1.2	1.3	1.6
**Tubular dilatation**	1.0	1.2	1.4	2.0	1.4
**Exfoliated cells**	1.0	1.0	1.1	1.0	1.6
**Leucocyte infiltration**	0.2	0	0.3	0.1	0.4
**Brush border loss**	1.0	1.2	1.1	1.3	1.3
**Total histopathologic score**	8.7	9.4	10.3	10.4	10.7

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
