# Peer review of "Beneficial Effects of Flaxseed and/or Mulberry Extracts Supplementation in Ovariectomized Wistar Rats"

_nutrients, 2022, doi:10.3390/nu14153238_

Round 1

Reviewer 1 Report

The manuscript described the beneficial effects of flaxseed and/or mulberry extracts in ovariectomized (low endogenous oestrogen) Wistar rats. Antioxidant activity of extracts, weight gain, pituitary weight, the predominance of vaginal epithelial cells, and lipid profiles, liver enzymes have been evaluated. The results suggested flaxseed and mulberry extract mitigate/prevent the negative effects caused by low estrogenic action.

Major questions:

1.       In introduction, it is hard to understand the materials used. It lacks explanation why author use flaxseed and mulberry combinations since these are totally two different medicinal plants.

2.       Author stated the beneficial effect due to their antioxidant activity. If there is any reference to confirm this or any curve or p value to show their relationship. I think in introduction, the author needs to clear explain, such readers could follow.

3.       The doses used in present study is 400 mg/kg (not clear, the concentration is to B.W. ?) and the combination is 200 mg/kg flaxseed+ 200 mg/kg mulberry.

Here there are two questions: 1), from the results, the body weight gained over 20 g in 60 days, it is a lot, did the author adjusted the concentration of extracts every one/two weeks, or just only prepared a stock at the beginning based on the body weight in first week (it seems the author did like this, but since the body weight is increasing, so the treatment dose is not 400 mg/kg B.W.))? I did not find any information in the methods part.

2), why the author chose 1:1 extract combination to treat rats, not other ratio?

Minor comments:

1.       In methods part, it is better to move “the flaxseed and mulberry extract preparation” to the first sub-section, since it is your material.

2.       In methods part, there are many several mistakes, such as lack of space between value and unit, such as line 121, 127, 158, 161, 167, 170. Also, please keep consistent always, sometimes full name (abbreviated name), such as line 153-154, but sometimes abbreviated name (full name), such as line 191….. there are many inconsistencies in the whole text, please keep consistent always.

3.       Line 153-154, such as lipid profile names, they are abbreviated here, but many terms have appeared in introduction, please check carefully.

4.       Line 179, 182, 192, please clarify 10%, 70%..., is v/v or w/w.

5.       Line 127, the control group gavage saline, but why Oestrogen was diluted in distilled water, not in saline?

6.       In results part, line 243, “IC50: half maximal inhibitory concentration (mg/mL) necessary for the antioxidant (sample) to reduce the initial DPPH radical of the reaction by 50%”, what is the meaning of this sentence?

7.       For the results analysis, why the author did not use ANOVA analysis to compare all groups?

8.       In results, Line 236, “while the flaxseed extract could not be quantified (Table 1)”. The author only simply explained in line 376 with a reference, but not clearly. Please explain it.

9.       In discussion, line 372-373, 44.77%, 66 and 76%, please keep consistent with the same decimal of values. Please check the whole text.

10.    Line 388-389, the author stated “Thus, the high concentrations of total phenolics and high anti-oxidant power found in the extracts used can be directly related to the positive results observed in animals without oestrogenic action”. How the author concluded it. How it was related, is there is a correlation curve and significant curve? how about the unsaturated fatty acids in flaxseed?

11.    Line 396-397, “Male rats supplemented with flaxseed meal showed less weight gain compared to the control group that did not receive supplementation”. What is flaxseed meal? How long is the supplementation?

12.    How long the rats were recovered from OVX when it begin supplementation, did the author measure oestrogen level before treatment?

Reviewer 2 Report

In the article “Beneficial effects of Flaxseed and/or Mulberry Extracts Supplementation in Ovariectomized Wistar Rats” the Authors examined flaxseed and mulberry extracts on rats. Moreover, they evaluated the antioxidant capacity of the natural substance using specific tests.

1.    Authors should write the brief description of the tests used in their article. It is annoying searching for each method described in other cited articles anytime I do not know it.

2.    The sentence “ it is necessary to understand the effects of certain medicinal plants rich in phenolic compounds and with antioxidant effects to enable the minimization of the effects caused by the reduction of oestrogen actions without demonstrating toxicity to the organism” does not represent the real aspects of this paper-I did not find any explanation how those substances act. I was searching for the in vitro assays in cell culture which can present the action of the extracts.

3.    Introduction part is very short.

4.    Material and methods part is very chaotic, Authors mixed mg-milligrams, numbers with the 1,2,3. There should be an explanation of why authors decided to do the test, what they want to measure and how.

5.    Why do Authors use the  ‘;’ ? I don’t understand the choice.

6.    English is extremely chaotic, many words are repeated, and sometimes I got the sentence hard to understand.

7.    Figure 1. Pituitary weight is presented in grams, but it should be presented in mg.

8.    The quality of Figure 2 pictures is very low, nothing can be observed. Authors should stain the cells to prove that the cells they marked are those they indicated in the text.

9.    The obtaining and preparing of the administered solutions should be well written and described so that other authors can repeat the method.

10.  Figure 5 presents very low quality.

11.  The discussion part presents others researchers' results, but without details in their studies. Therefore, it is impossible to compare the results gained by the Authors, because we do not know if there were studies on cells, on rats, or another way round. Moreover, the Authors cited results that were done on males, other species or with other plant extracts, thus it is hard to discuss results and the possible mechanisms.

Reviewer 3 Report

Using ovariectomized animal models, the authors investigated the phytoestrogenic effects of flaxseed and/or mulberry extracts supplementation for improving climacteric disturbance. Like administering estrogen, flaxseed and/or mulberry extracts reduced weight gain, increased pituitary weight, increased vaginal epithelial cell growth by their predominance in the virginal lavage, improved serum lipid levels, and increased endometrial thickness. On the other hand, these extracts had no adverse effect on the liver and kidney. Taken together, they conclude that flaxseed and mulberry extracts may bring benefits for mitigating the negative effects caused by lowering estrogen during menopause and postmenopausal.

The reviewer thinks that the experiments were performed carefully, and the results corresponded to their conclusions. On the other hand, the introduction seems insufficient to present the study's background, including why phytoestrogenic compounds were focused on, why they paid attention to flaxseed and mulberry extracts, what is known/unknown regarding phytoestrogenic action of these extracts, and why they performed a combination of both extracts. In addition, the purpose of the study seems not to be written legibly (Lines 67-71). Don’t they want to show beneficial phytoestrogenic action of the extracts minimizing the adverse effects in the animal models of menopause? Furthermore, the description of the discussion section seems to be redundant. Repetition of the results also exists. Revising the discussion section with compacting quotation parts and reinforcing their results with the previous reports would be preferred. Limitations of the study would also be added for further study to use these extracts for clinical purposes in humans.

Round 2

Reviewer 1 Report

The authors have addressed some of my questions, but not all. I would like the authors clarify these two questions before acceptance.

1.       For my question, whether the author adjusted the concentration of extracts according to the gain of body weight (gain over 20 g), the author replied they adjusted the volume of the administration of the extracts if necessary. What is the meaning of if necessary? How do you clarify it is necessary? Is there a criterion?

2.       I asked why the author chose 1:1 extract combination. The author replied to leave the same final concentration for all groups (400 mg/kg B.W.). The ratio is different with the final concentration, I mean why the author use 200 mg/kg B.W. + 200 mg/kg B.W., but did not use 100 mg/kg B.W.+300 mg/kg B.W. or others? What is the logic here?

3.       There are still small mistakes along the text, please check.

Author Response

Reviewer 1- Nutrients

July 13th, 2022

Re: Response round 2 of manuscript ID nutrients-1780740

Dear Reviewer 1

We are very pleased that our answers were accepted in almost their entirety. Our answers to clarify any doubts that may remain are given below:

The authors have addressed some of my questions, but not all. I would like the authors clarify these two questions before acceptance.

  1. For my question, whether the author adjusted the concentration of extracts according to the gain of body weight (gain over 20 g), the author replied they adjusted the volume of the administration of the extracts if necessary. What is the meaning of if necessary? How do you clarify it is necessary? Is there a criterion?

The volume adjustment was made if the animal had increased weight, as there were animals that gained weight each week but had no weight change in another week. As a result, there were weeks when it was necessary to change the administration volume, but in another week, it was not necessary to adjust the volume again because the weight gain was not significant. For example, animals that gained around 20g of body weight at the end of the experiment did not have major changes in the volume administered, as the dosage was 400 mg/kg B.W., and if at the beginning of the experiment the animal was weighing 210g and received 84mg/day of extract in a volume around 450uL, this animal with 20g more at the end of the experiment was receiving a final volume of around 490uL at the end of the experiment, always maintaining the dosage of 400mg/kg B.W.

  1. I asked why the author chose 1:1 extract combination. The author replied to leave the same final concentration for all groups (400 mg/kg B.W.). The ratio is different with the final concentration, I mean why the author use 200 mg/kg B.W. + 200 mg/kg B.W., but did not use 100 mg/kg B.W.+300 mg/kg B.W. or others? What is the logic here?

The concentration of substances present in one extract is different from the concentration in the other, however, as we used a dosage of 400 mg/kg of body weight of flaxseed extract in one group and the same dosage for mulberry extract in another group, our idea was to verify if half of that dosage of flaxseed extract plus half of the dosage of mulberry extract in a single group, could bring some benefit or different results from the other groups analyzing. It’s because, for example, the extracts used have high phytoestrogens concentrations, however, flaxseed is rich in lignans, while mulberry is rich in isoflavones, substances that have different affinities for estrogen receptor subtypes and therefore the idea of ​​using a solution composed of the two extracts, in the same dosage (200mg/kg B.W.) of each extract and that the final dosage was the same as the other groups that received flaxseed or mulberry extract supplementation. Therefore, the choice to use 200 mg/kg B.W. + 200 mg/kg B.W., and no other proportions, since could lead to the conclusion that the effect that had been observed would be more due to the fact of having used a higher dosage of one extract than the other and not due to possible synergism of both extracts. The reason for this choice was to verify if the use of both extracts could bring more benefits than a higher dosage of just one extract.

Reviewer 2 Report

Dear Authors,

I am impressed with the work that you did to improve the manuscript.

I do not have any more comments.

All the best,

M